



# High resolution (1 km) satellite rainfall estimation from SM2RAIN applied to Sentinel-1: Po River Basin as case study

Paolo Filippucci[1,2,3], Luca Brocca[1], Raphael Quast[2], Luca Ciabatta[1], Carla Saltalippi[3], Wolfgang Wagner[2], Angelica Tarpanelli[1]

[1]National Research Council, Research Institute for Geo-Hydrological Protection, Perugia, Italy
[2]TUWien (Technische Universität Wien), Department of Geodesy and Geoinformation, Wien, Austria
[3]Università degli Studi di Perugia, Department of Civil and Environmental Engineering, Perugia, Italy

*Correspondence to*: Paolo Filippucci (paolo.filippucci@irpi.cnr.it)

**Abstract.** Satellite sensors to infer rainfall measurements have become widely available in the last years, but their spatial
resolution usually exceed 10 kilometres, due to technological limitation. This poses an important constraint on their use for application such as water resource management, index insurance evaluation or hydrological models, which require more and more detailed information.

In this work, the algorithm SM2RAIN (Soil Moisture to Rain) for rainfall estimation is applied to a high resolution soil moisture product derived from Sentinel-1, named S1-RT1, characterized by 1 km spatial resolution (500 m spacing), and to the 25 km
ASCAT soil moisture (12.5 km spacing), resampled to the same grid of S1-RT1, to obtain rainfall products with the same spatial and temporal resolution over the Po River basin. In order to overcome the need of calibration and to allow its global application, a parameterized version of SM2RAIN algorithm was adopted along with the standard one. The capabilities in estimating rainfall of each obtained product were then compared, to assess both the parameterized SM2RAIN performances and the added value of Sentinel-1 high spatial resolution.

The results show that good estimates of rainfall are obtainable from Sentinel-1 when considering aggregation time steps greater than 1 day, since to the low temporal resolution of this sensor (from 1.5 to 4 days over Europe) prevents its application to infer daily rainfall. On average, the ASCAT derived rainfall product performs better than S1-RT1 one, even if the performances are equally good when 30 days accumulated rainfall is considered, being the mean Pearson's correlation of the rainfall obtained from ASCAT and S1-RT1 equal to 0.74 and 0.73, respectively, using the parameterized SM2RAIN. Notwithstanding this, the
products obtained from Sentinel-1 outperform those from ASCAT in specific areas, like in valleys inside mountain regions and most of the plains, confirming the added value of the high spatial resolution information in obtaining spatially detailed rainfall. Finally, the parameterized products performances are similar to those obtained with SM2RAIN calibration, confirming the reliability of the parameterized algorithm for rainfall estimation in this area and fostering the possibility to apply SM2RAIN worldwide even without the availability of a rainfall benchmark product.



## 1 Introduction

Water supplies are not endless. Water consumption has been steadily increased in the last century (Kummu et al., 2016) and the current climatic crisis is expected to further increase the water intake: the water availability should reduce, while irrigation demand should increase. Many areas will experience water scarcity due to this phenomenon, as it is already happening (Rockström et al., 2012). In this framework, water resource management is extremely important to increase conservation and use efficiency of this precious resource. Spatially detailed measurements of the various water cycle components are therefore needed to stakeholders and companies involved in water management, in order to increase the intervention capacities and reduce wastage. High resolution information are also required to improve hydrological models performances and capabilities, whose need of high quality input data with sufficient resolution characteristic is increasing along with the models complexity (Silberstein, 2006; Ragettli et al., 2013). Insurance companies are also demanding high spatial resolution data, even at monthly temporal scale, with the purpose to develop index-based insurance for small-scale agricultures (Enenkel et al., 2019; Black et al., 2016). One of the most important variables for these objectives is precipitation, indicated by Global Observing Systems Information Center (GCOS) as an Essential Climate Variables (ECV), i.e., a variable whose knowledge is needed in order to understand the evolution of the climate, to assess the related risks and to develop mitigation and adaptation strategies. Measurements of rainfall, the liquid fraction of precipitation, are traditionally obtained from raingauge sensors, which are characterized by a high degree of precision (La Barbera et al., 2002). Notwithstanding this, the rainfall spatial variability makes the current raingauge network inappropriate to describe in detail its distribution over the full globe since the number of gauges is too scarce with respect to Earth surface and they are unequally distributed, being the majority of them located in the most developed countries (Villarini et al., 2008; Kidd et al., 2017; Dinku, 2019).

In this framework, satellite rainfall estimates have demonstrated their potential to support, integrate and in some cases substitute ground-based networks (Barret and Beaumont, 1994; Kidd and Levizzani, 2011). Historically, two main approaches are adopted to estimate rainfall from space: the traditional "top-down" approach, where the instantaneous precipitation rate is estimated from the clouds upwelling radiation or the rain drops backscatter sensed by radar and/or radiometers, and the more recent "bottom-up" approach, where the rainfall rate over land is inferred by Soil Moisture (SM) observations. The peculiarity of the last approach lies in its capacity to estimate the accumulated (not instantaneous) rate by using the soil as a "natural raingauge" (Brocca et al., 2014). Among the algorithms that use this approach, SM2RAIN has distinguished itself for its versatility and simplicity. By inverting the soil water balance equation, this algorithm allows the estimation of the rainfall occurred between two SM measurements. It has already been applied worldwide to both regional (Tarpanelli et al., 2017) and global (Ciabatta et al., 2018; Brocca et al., 2019) satellite SM products, obtaining satisfactory results, in particular over regions characterized by scarce raingauge density (Massari et al., 2020).

Nevertheless, the major limitation of satellite observations, regardless the adopted approach, is the inherent "technological" compromise between temporal and spatial coverage: SM and rainfall satellite products are usually characterized by a frequent revisit time (<1 day) and a coarse spatial resolution (~10-50 km). It is of primary importance to obtain data with high temporal



and spatial resolution, in order to enhance the prediction capability of hydrological models requiring high resolution input data (Merlin et al., 2008) and the spatial accuracy of the information related to water resource. The first attempt to accomplish the

objective of high spatial resolution was the use of downscaling procedures. Many different approaches, from geostatistical analysis to data fusion, have been developed in the last years in order to obtain sub-pixel information from coarse resolution products (Peng et al., 2017) to be used in different applications (e.g. Dari et al., 2020). However, their results were often unsatisfactory, because of the limitations of the auxiliary data (e.g., cloud cover for optical data and model errors when using model data) and the uncertainties of downscaling algorithms (Peng et al., 2017).

Recently, the new launched Sentinel Missions of the European Earth Observation program Copernicus, has opened new possibilities to overcome these issues. Specifically, Sentinel-1 (S1) mission is composed of two satellites that share the same orbit 180° apart and follow a strict acquisition scenario with a 12-days repeat cycle (6-days by considering both satellite), each carrying an identical C-band Synthetic-Aperture Radar (SAR) sensor capable to sense high resolution microwave backscatter (down to 5 meters). This setup leads to a revisit frequency of 1.5-4 days over Europe, thanks to the overlap of the orbits. The

condition of high spatial and medium temporal resolution is, for the first time, met by the two S1 satellites currently in orbit. SM measurements with 1 km spatial resolution can be obtained from this mission (Bauer-Marschallinger et al., 2018).

The application of SM2RAIN to such data could therefore allow to obtain high resolution (1 km) rainfall estimates over land. This approach, however, is limited by the need to calibrate SM2RAIN algorithm against a rainfall product with spatio-temporal characteristic similar to those of the input SM. Datasets with such spatio-temporal characteristics are rarely available, due to

the already mentioned scarce density of ground-based networks, thus limiting the calibration and validation of high resolution rainfall from SM2RAIN only over few selected areas. This issue can be overcome by exploiting the parameterized version of SM2RAIN that has been found capable to estimate SM2RAIN parameter values, by accepting a limited reduction of the performance, only from the knowledge of SM noise, the topographic complexity and each rainfall climatology, without the need of a calibration procedure (Filippucci et al., 2021).

In this work, the parameterized and calibrated versions of SM2RAIN algorithm were applied to a SM product derived from Sentinel-1 over the Po River basin in northern Italy, with the scope of evaluating their capabilities in reproducing high resolution rainfall (1 km). The product, from here on named S1-RT1, was obtained by using a retrieval algorithm based on the first order solution of Radiative Transfer equation (RT1, Quast et al., 2019), and is characterized by 500 m spatial sampling and 1.5-4 days temporal resolution. Po River basin area was selected because a ground-based rainfall dataset with 1 km spatial

resolution and 1 hour temporal resolution is available over the area, thanks to the fusion of raingauges and weather radar measurements through the Modified Conditional Merging (MCM) algorithm (Bruno at al., 2021). Furthermore, Po River basin comprehends many geographical features, such as mountains, hills, lakes, rivers and plains, which make it a good test area for this analysis. Both SM2RAIN versions were applied also to ASCAT-derived SM, after it was regridded to S1-RT1 coordinates, in order to assess the benefits derived from the use of high resolution SM by comparing the resulting rainfall products

performances.





The paper is structured as follows: the study area and the data collected for this study are presented in Section 2, the two SM2RAIN versions and the selected performance scores are described in Section 3. The obtained results and the spatial distribution analysis are shown in Section 4. Finally, the conclusions of the analysis are summarized in Section 5.

## 2 Study area & Data

**2.1 Study area**

The analysis was conducted over the Po River Basin, located in Northern Italy (Fig. 1). The basin extends from the Western Alps to the Adriatic Sea, including Italian and Swiss territories. The region covers an area of around 71000 km2: the Alps outline the boundaries of the basin to the North and West, with altitudes up to 4809 m, while the Apennines mark the South borders. The Po Plain extends to the central part of the basin, broadly divided into a northern and a southern section: the former

is generally unsuitable for agriculture, while the latter is more fertile and well irrigated. The average annual precipitation goes from ~700 to ~1500 mm/year in the analyzed period, 2016-2019, equally distributed during the year, with maximums occurring during autumn and spring seasons. Po basin area is classified as Cfa (Temperate climate, without dry season and with hot summer) by the Köppen-Geiger climate classification (Peel et al., 2007). In this study, the Swiss fraction of the Po River basin (red area in Fig. 1) was excluded from the analysis due to the unavailability of raingauge data.

**2.2 Data**

Several datasets were collected in this study to analyse the feasibility of high resolution rainfall estimations from SM2RAIN. Specifically, SM products from ASCAT and S1 sensors were analysed, alongside with the selected benchmark rainfall dataset MCM and the data needed for the parameters estimations within the parameterized SM2RAIN, i.e., SM noise from ASCAT, topography and rainfall climatology.

*SM measurements*

SM data at 25 km spatial resolution (12.5 km spacing) were obtained from ASCAT, while the high resolution 1 km estimates (500 m spacing) were derived from the application of S1-RT1 algorithm to Sentinel-1 data (Quast et al., 2019).

ASCAT is an active microwave sensor that measures backscatter radiation at 5.255 GHz (C-band) mounted on MetOp-A (launched 19/10/2006), MetOp-B (launched 17/09/2012) and MetOp-C (launched 07/11/2018) satellites. The conjunct use of

multiple satellites allows to achieve sub-daily estimates of relative SM, i.e., the soil moisture saturation fraction, over most of Earth (Wagner et al., 2013). The SM data, together with the associated noise, were downloaded from EUropean organisation for the exploitation of METeorological SATellites (EUMETSAT) Satellite Application Facility on Support to Operational Hydrology and Water Management (H SAF) H115 and H116 products, comprehending data from both MetOp-A and MetOp-B, within the period 2016-2019. Surface state information is available with the dataset, therefore data marked as "frozen" were

discarded from the analysis.



Sentinel-1 mission is composed by a constellation of two polar-orbiting satellites, Sentinel-1A (launched 03/04/2014) and Sentinel-1B (launched 25/04/2016), sharing the same orbital plane 180° apart, each carrying a single C-band Synthetic Aperture Radar (SAR) instrument operating at a center frequency of 5.405 GHz. S1 sensors can operate in four exclusive imaging modes with different spatial resolution (down to 5 m) and swath width (up to 400 km). Particularly, the Interferometric

Wide (IW) swath mode, the main sensing mode over land, offers a 20 m x 22 m spatial resolution with a 250-km swath. The revisit time of a single satellite is 12 days, which reduces down to 6 days when considering both sensors. Nevertheless, the acquisition strategy prioritizes European landmasses over other regions, therefore the effective temporal resolution over Europe is between 1.5 and 4 days by taking advantage of the overlapping ascending and descending orbits.

SM retrievals at 1 km spatial resolution were obtained by applying a first-order radiative transfer model (RT1) (Quast et al.,

2019) to a 1 km Sentinel-1 backscatter ($\sigma0$) dataset sampled at 500 m pixel spacing (Bauer-Marschallinger et al., 2021). RT1 is based on a parametric (first-order) solution to the radiative transfer equation (Quast and Wagner, 2016) in conjunction with a timeseries-based non-linear least squares regression to optimize the difference between (incidence-angle dependent) measured and modelled $\sigma0$. The scattering characteristics of soil- and vegetation are modelled via parametric distribution functions, and the relative SM content (scaled between 0 and 1) is found to be proportional to the nadir hemispherical

reflectance (N) of the bidirectional reflectance distribution function used to describe bare-soil scattering characteristics.

To correct for effects induced by seasonal vegetation dynamics, scaled Leaf Area Index (LAI) timeseries provided by ECMWFs ERA5-Land reanalysis dataset have been used to mimic the temporal variability of the vegetation optical depth, accounting for the attenuation of the radiation during propagation through the vegetation layer. Remaining spatial variabilities in soil and vegetation characteristics are accounted for by the model-parameters "single scattering albedo" ($\omega$) and soil-

scattering directionality (ts). Within the retrieval-procedure, a unique value for N is obtained for each timestamp, alongside a temporally constant estimate for ts and an orbit-specific estimate for $\omega$ for each pixel individually based on a 4-year timeseries from 2016-2019.

A detailed description and performance-analysis of the used soil-moisture dataset will be given in Quast et al., in preparation. Due to the presence of systematic differences between Sentinel-1 acquisitions from different orbits, the obtained soil-moisture

timeseries exhibits a periodic disturbance, attributable to unaccounted differences in soil- and vegetation characteristics with respect to the different viewing-geometries. To correct these systematic effects, the timeseries are split with respect to the Sentinel-1 orbit ID and normalized individually to a range of (0, 1) prior to the incorporation into the SM2RAIN algorithm.

In order to obtain data with the same time spacing, SM data were linearly interpolated at midday and midnight for both datasets. If no data were found within 5 days, each datum in the interval was set to Not a Number (NaN). ASCAT data were resampled

on S1-RT1 grid using a weighted average of the four nearest pixels, to allow the inter-comparison of the data. Finally, all the SM products were masked for frozen soil and snow cover conditions, by downloading the Soil Temperature (Tsoil) of the first soil layer (0-7 cm) and Snow Depth data from ERA5-Land (see description below), and excluding the SM estimates obtained over pixels showing a Tsoil < 2 °C or a snow depth > 0.01 m.

*Rainfall measurements*



Two rainfall datasets were considered, to be used as benchmark for the performance assessment and as input for the parameterized version of SM2RAIN, respectively. The first one is a product derived from the integration of ground radar and raingauge measurements over the Italian territory through the MCM algorithm (Bruno et al., 2021). Indeed, a dense network of raingauges and weather radars is available over the Italian territory, making possible to obtain hourly measurement of rainfall in real-time. While raingauges allow a good estimation of point rainfall, radar measurements give a good estimation

of the general covariance structure of rainfall. MCM uses radar data to condition the spatially limited information of raingauges, generating a rainfall field with a realistic spatial structure and constrained to raingauge values. The resulting rainfall product is characterized by high spatial (1 km) and temporal (1 h) resolution. These attributes make it a suitable choice for the purpose of comparison with SM2RAIN estimations from high resolution SM. In this work, the MCM hourly information were resampled to S1 data coordinates. MCM data were temporally accumulated at 12 hours, obtaining two cumulated rainfall

measurements per day, respectively at midday and midnight. Rainfall measurements greater than a threshold of 800 mm/day were considered not valid and discarded from the analysis. Even if MCM data were available for the full Po River basin, the territories outside the Italian boundaries were excluded from the analysis due to the absence of raingauges data.

In order to apply the parameterized version of SM2RAIN (see section 3.2), the mean daily rainfall of each pixel in the study area is needed and it was obtained by downloading Total Precipitation and Snowfall daily measurements from the European

Centre for Medium-Range Weather Forecasts (ECMWF) Reanalysis 5th Generation Land product (ERA5-Land) for the period 1981-2021. ERA5-Land provides estimation of various climate components combining models with observations (Hersbach et al., 2020). The original ERA5 spatial resolution is around 30 km, resampled on a regular 25 km grid. ERA5-Land was produced by regridding the land component of the ECMWF ERA5 climate reanalysis to a finer spatial resolution (0.1-degree). Daily rainfall data were obtained by subtracting the Snowfall component from ERA5-Land Total Precipitation. The obtained

rainfall data were then regridded on S1 grid using a weighted average of the four nearest pixels, as done with ASCAT SM data. The 30-year averaged mean daily rainfall was then calculated for each pixel.

*Topography measurements*

Elevation data from Terra Advanced Spaceborne Thermal Emission and Reflection Radiometer (ASTER) global Digital Elevation Model (DEM) Version 3 (ASTGTM) were downloaded. The product provides altitude land data at a spatial

resolution of 1 arc second (~30 meters resolution at equator). In order to obtain the topographic complexity of each S1 pixel, the standard deviation of the DEM values within each pixel was calculated.

Data interpolation and regridding are expected to introduce small-scale noise in the datasets. Notwithstanding this, the interpolation is unavoidable in order to analyse all the products with the same spatial and temporal sampling.


## 3 Methods

### 3.1 SM2RAIN


The algorithm adopted to estimate the rainfall accumulated between two consecutive SM measurements was SM2RAIN, developed by Brocca et al. (2013; 2014) by inverting the soil water balance equation. This can be described by:

$$Zn \, {dSM(t)}/{dt} = p(t) - r(t) - e(t) - g(t) \tag{1}$$

where $Z$ [mm] is the depth of the considered layer, $n$ [m³/m³] is the soil porosity, $SM(t)$ is the relative SM [-], $p(t)$ is the

rainfall rate [mm/d], $r(t)$ is the surface runoff rate [mm/d], $e(t)$ the evaporation rate [mm/d] and $g(t)$ the drainage rate [mm/d]. Eq. (1) can be simplified as

$$p(t) = Z^* \, {dSM(t)}/{dt} + g(t) \tag{2}$$

during rainfall events, by considering evaporation and surface runoff negligible under this circumstance (Brocca et al., 2015) and $Z^* = Zn$. Finally, by expressing the drainage rate according to Famiglietti and Wood (1994) relationship, SM2RAIN

equation can be obtained:

$$p(t) = Z^* \, {dSM(t)}/{dt} + a \, SM(t)^b \tag{3}$$

where $a$ [mm/d] is the saturated hydraulic conductivity and $b$ [-] is the exponent of Famiglietti and Wood equation. In order to take into account for the low depth sensitiveness of satellite SM (few centimetres) and the signal noises, an exponential filter (Wagner et al., 1999; Albergel et al., 2008) is applied to the data before the application of SM2RAIN algorithm. In this

study, we adopted a modified exponential filter in which the filter characteristic time length, $T$, varies with SM, decreasing when SM increases according to a 2-parameters power law (Brocca et al., 2019). These 2 parameters are therefore needed along with $Z^*$, $a$ and $b$ to obtain the estimation of the rainfall between two consecutive SM measurements. In the standard SM2RAIN application, the 5 parameters are obtained through calibration against a reference rainfall dataset with similar spatial and temporal resolution, by minimizing the Root Mean Square Error (RMSE) between the estimated and reference data. The

calibrated SM2RAIN has already been applied to different satellite and in situ SM dataset (Ciabatta et al., 2018; Brocca et al., 2019; Filippucci et al., 2020) showing good performance, particularly over poorly gauged regions (Massari et al., 2020).

### 3.2 Parameterized SM2RAIN

Filippucci et al. (2021) developed four parametric relationships that allow to obtain the SM2RAIN parameters along with the $T$ parameter of the original exponential filter (not the modified version above adopted), without calibration. $T$, $Z^*$, $a$ and $b$ can

be therefore obtained from the knowledge of SM timeseries and its noise, the topographic complexity and the mean daily rainfall of the standard year, obtained averaging the rainfall in the same Day of Year (DOY). In particular:



$$T = 0.8351 + 1.2585\ \overline{SMnoise}\ std(|SM_d|) + 0.2777\ {std(|SM_d|)}/{\overline{P}}\ topC \tag{4}$$

where $\overline{SMnoise}$ is the average SM noise in the considered pixel, $(|SM_d|)$ is the standard deviation of the absolute values of the SM temporal variations, $\overline{P}$ is the pixel mean daily rainfall and $topC$ is the topographic complexity.

After the calculation of $T$ and the application of the exponential filter to the SM timeseries, it is possible to calculate the remaining SM2RAIN parameters according to:

$$Z^* = 10.0678 + 0.5350\ {\overline{P}}/{\overline{|SM_{fd}|}} \tag{5}$$

$$a = -1.3177 + 13.3579\ Z^*\ \overline{|SM_{fd}|} \tag{6}$$

$$b = 3 + {2}/{0.4118} + 0.324 * \log a \tag{7}$$

where $\overline{|SM_{fd}|}$ is the average of the absolute values of the filtered SM temporal variations. The coefficients of the equations above are slightly different from those published on Filippucci et al. (2021), in which the Digital Elevation Model (DEM) adopted to obtain the pixels $topC$ had a spatial resolution of 5 arc minutes, unsuitable for the current analysis. Therefore, the parametric relationships were recalculated by substituting the previous ETOPO5 DEM information with ASTGTM DEM, repeating the same steps of Filippucci et al. (2021).

**3.3 Performance scores**

In order to assess the performance of the rainfall estimates obtained from SM2RAIN, different continuous metrics indices were calculated in comparison with the reference dataset, MCM. Specifically:

- Linear Pearson's Correlation ($R$), that is an index to express the linear relationship between two set of data. Its value ranges between -1 and +1, where -1 indicate perfect negative linear relationship, +1 means perfect positive linear relationship and 0
means no statistical dependency.

- *BIAS*, index that measures the systematic over- or under-estimation of one dataset with respect to the reference data. In this paper, it is calculated as the difference between the estimated and the observed rainfall: therefore, negative BIAS values indicate a systematic rainfall underestimation, while positive BIAS values mean the opposite.

- Root Mean Square Error (*RMSE*), that is widely used to measure the differences between two population values because it
takes into account three different sources of error together: decorrelation, BIAS and random error. It can be obtained by calculating the square root of the quadratic mean between single measurements of two datasets.





## 4 Results

### 4.1 Rainfall validation

In order to obtain rainfall measurements from the SM datasets, SM2RAIN algorithm was applied to both ASCAT and S1-RT1

SM products by using both the calibrated and parameterized versions. In the calibrated SM2RAIN, the algorithm parameters were estimated by minimizing the RMSE with respect to MCM rainfall product at daily time scale for both ASCAT and S1-RT1 SM. For the parameterized SM2RAIN version, the algorithm parameters were obtained through the parametric relationships developed by Filippucci et al. (2021), as mentioned above. Since no information regarding S1-RT1 SM noise was available, ASCAT SM noise characteristics were used to calculate S1-RT1 SM2RAIN parameters, assuming that since

both ASCAT and S1 sensors operate in C-band, the noises affecting the two SM products are similar. Indeed, the noise level of S1-RT1 is expected to be higher than ASCAT one. This sub-optimal configuration can be therefore considered as a first step to test the data: better results should be obtained when more accurate noise information will be available.

The obtained rainfall can be then accumulated at the desired time step. In order to consider the different temporal resolution of the selected SM products, sub-daily for ASCAT and from 1.5 to 4 days for S1, three accumulation time steps were chosen:

1 day, 10 days and 30 days. The daily rainfall was calculated only for ASCAT product, since the low temporal resolution of S1 prevents to obtain significant results at such temporal step.

Figure 2 shows the average 30 days rainfall obtained by the application of the parameterized SM2RAIN to ASCAT and S1-RT1 SM products. By comparing the two figures, the improved resolution of the rainfall obtained by applying SM2RAIN to S1-RT1 SM with respect to its application to ASCAT is evident: the higher spatial resolution of S1-RT1 allows the generation

of detailed features, even if with a granular effect likely due to the uncertainties of the measurements, and with patterns related to the spatial variation of S1 temporal resolution (compare with Fig. 4e).

The results in terms of R, RMSE and BIAS are shown in Fig. 3, considering the chosen time steps. In order to maximize the reliability of the obtained performances, the rainfall accumulation was carried out by summing up only the data that were available in both the SM2RAIN estimations and the benchmark, for each SM2RAIN product separately. In this way S1-RT1

performances can be better assessed, since a direct accumulation would penalize this product due to the long period of no-data caused by S1 low temporal resolution.

The SM2RAIN product obtained from ASCAT allows to well reproduce the rainfall of the Po River basin at daily time scale thanks to the high temporal resolution of ASCAT (sub-daily frequency), with a median R equal to 0.61 when the parameterized product is considered and to 0.64 in case of calibration, confirming the good quality of the data and the importance of its

temporal resolution. At higher aggregation time steps, the performances of ASCAT-derived rainfall improve, being the median R by the parameterized (calibrated) SM2RAIN equal to 0.71 (0.75) for the 10 days accumulated rainfall and to 0.74 (0.77) when 30 days accumulation is considered. Good results are also obtained from the application of SM2RAIN to S1-RT1, with a median R of 0.61 (0.65) and 0.73 (0.75) at 10 and 30 days time step, respectively. Albeit ASCAT-derived rainfall performs better than the one from S1-RT1 at 10 days, they are equally good for the 30 days accumulated rainfall. The results also confirm





the good capabilities of the parameterized SM2RAIN algorithm in rainfall estimation, considering the small differences
between the performances obtained by the two algorithm versions, in particular when 30 days are considered. The only
exception is the BIAS index, which, as expected, is significantly larger in the parameterized products with respect to the
calibrated ones. The increased BIAS is due to the ERA5-Land data used to obtain the climatology of the area, being its spatial
resolution much lower than the one adopted for this study (i.e., 1 km) and the average pattern of rainfall quite different from
the one measured by MCM.

## 4.2 Spatial validation of rainfall products

Even if the ASCAT product (with lower spatial resolution) is on average the best performing, the spatial comparison of the
performances is important to understand the added value of high resolution SM. In order to better evaluate the differences
between the rainfall estimated from ASCAT and S1-RT1, the Pearson's correlation performances of the 30 days accumulated
rainfall derived from the two SM products are analysed in this section. This temporal step was selected since it is suited for a
quality comparison of the two products, being less influenced by the different temporal resolution of the sensors, and because
it is optimal for agricultural application.

Generally good performances are obtained from both rainfall products, as shown in Fig. 4a and 4b. Some areas with low R
values are shared by both ASCAT and S1-RT1 derived rainfall products. Over mountain areas the errors are mostly related to
the lower accuracy of C-band SM data, due to shadowing effects and layover (a distortion that occurs in radar imaging when
the signal reflected from the top of a tall feature is received by the emitter before the one of the base, ULABY ET AL., 1981).
The presence of water bodies at the river outlet and over the paddy fields in the western part of the Po basin is also affecting
SM, and hence rainfall, retrieval accuracy. Finally, the yellow "holes" in the correlation maps resemble the errors caused by
low quality gauge data, which affect the rainfall estimation of the surrounding of the gauge sensor. It should also be noticed
that many low performing areas are located close to urban centers, which may affect both the SM retrieval quality and the
raingauge measurements (see also Appendix-A). Notwithstanding this, it is impossible to remove the alleged "bad" gauge
stations from the benchmark product, being MCM an operative product and the clear identification of these stations often
challenging.

The spatial comparison between the performances of the ASCAT and S1-RT1 derived rainfall is instead shown on Fig. 4c,
displaying the difference between the correlation values of the two products. Red area means that the S1-RT1 product is
performing better, whereas blue areas highlight where ASCAT is providing more accurate rainfall estimates. First of all, it
should be noted that while ASCAT derived rainfall product shows average correlation values over the mountainous region in
the North and West of the map, S1-RT1 correlation are alternatively extremely low or extremely high. This important
difference is caused by the high spatial resolution of S1-RT1 product: the improved resolution permits to clearly distinguish
the "good" signal originated from the valleys and the "bad" signal coming from the mountain slopes, affected by the noise
generated from the aforementioned shadowing effects and layover. This distinction leads to generating areas with respectively
very good (valleys) and very bad (mountains) rainfall estimations. ASCAT spatial resolution, instead, does not permit to





distinguish the signals of the two geographical features that therefore overlap, causing lower performances over the valleys and higher performances over the slopes in comparison with S1-RT1. The low performances of the pixels located over the mountain slopes are also responsible of the long violin plots tails of S1-RT1 performances that can be noticed in Fig. 3. S1-RT1 results are particularly lower than those from ASCAT due to the fact that S1-RT1 product calibration was carried out without considering any snow masking, thus reducing the quality of the solution in the pixels affected by snow cover.

A smaller difference in performance can be noticed over the plain, in particular in the north-eastern section, where S1-RT1 rainfall performs overall better than ASCAT. Conversely, in the southern section and specifically over the areas surrounding the Po River and its tributaries, ASCAT derived rainfall is better than S1. An explanation of this behaviour can be found in the intensive irrigation practice over this area. Irrigation events cause an increase of the fields SM (Filippucci et al., 2020) that should be sensed by satellites sensors. However, the area surrounding the Po River is composed by many small fields (few hectares each) managed by different farmers, where the irrigation timing is not concurrent. ASCAT sensor is not able to distinguish the resulting irrigation signal (Brocca et al., 2018), because of its low spatial resolution (25 km) that cause the signals of each field to overlap and average with each other. S1, instead, is more sensitive to the irrigation signal, thanks to its higher spatial resolution.

Considering that the rainfall benchmark product does not contain irrigation information, the drop in Pearson's correlation of the S1-RT1 derived rainfall with respect to ASCAT could be related to the sensitiveness of the former to the aforementioned irrigation events, and not to the SM signal quality. It could be an additional information of great scientific interest but, unfortunately, the absence of detailed irrigation data for the Po Valley makes difficult to verify this hypothesis.

Finally, it should be also noted that this analysis could be biased in the areas characterized by a high presence of missing values (NaN) for one product with respect to the other, which hampers the statistical significance of the performance indices. Notwithstanding this, the absence of patterns in the maps that resemble the NaN distribution percentage shown in Fig. 4d and 4e, fosters the validity of the analysis.

The performance comparison in terms of RMSE and BIAS indices and the ones related to the calibrated SM2RAIN is here omitted for the sake of brevity, because no relevant additional information can be obtained from it.

In Fig. 5 and 6, the rainfall and SM timeseries of two pixels selected in the north-west of the Po basin are shown, as example of the increased capacity of S1-RT1 for rainfall retrieval in the mountainous area. Winter and early-spring measurements are masked in both pixels, due to frozen condition or snow cover, according to ERA5-Land information. The pixel in Fig. 5 is selected over one of the mountain valleys of the Italian territory (7.152°E, 45.710°N), inside the Italian region Valle d'Aosta, in order to show how S1 spatial resolution increase the capabilities in rainfall estimation over such region. By observing the rainfall timeseries in Fig. 5a and the standard month distribution in Fig. 5b, it can be noted how S1-RT1 derived rainfall is in better accordance with the observed one, in particular during autumn months. During late spring and summer, S1-RT1 and ASCAT estimates are more similar, with S1-RT1 that often underestimates the observed rainfall, also with respect to ASCAT. In Fig. 5d, the same behaviour can be noted on the averaged SM trends, with the SM sensed by S1-RT1 being averagely less





than the one from ASCAT during late spring-summer and greater during the autumn season, probably due to the additional vegetation correction operated within S1-RT1.

## 5 Conclusion

Rainfall measurements from space have been lately widely used to increase the rainfall distribution knowledge and allow to
improve water resource management capabilities, but their spatial resolution is limited due to technological limitation. In this work, SM2RAIN algorithm was applied to a 1 km spatial resolution Soil Moisture (SM) product from Sentinel-1 (S1) obtained through an algorithm based on the first order solution of the Radiative Transfer equation, RT1, over the Italian fraction of the Po River Basin, to obtain an equal high resolution rainfall product from satellite remote sensing. This region was selected due to the availability of a benchmark dataset from radar and raingauge data, obtained through the Modified Conditional Merging
(MCM) algorithm. Two versions of SM2RAIN were applied in this analysis to compare the resulting performances: one uncalibrated, to foster the high resolution rainfall estimation in other regions where benchmark data are unavailable, and one calibrated against the observed data. In order to assess the improvements related the high spatial resolution of S1, SM2RAIN was also applied to ASCAT SM, resampled to S1-RT1 grid for comparison. The analysis was carried out at different temporal accumulation steps to take into account the different temporal resolution of the two SM products, 1.5 to 4 days for S1-RT1
and sub-daily for ASCAT, thus calculating the rainfall accumulated at 1 day, 10 days and 30 days.

The results show that is indeed possible to obtain high resolution rainfall data from S1, even if the low temporal resolution of the data does not allow to calculate daily rainfall. It is instead possible to calculate it with ASCAT data due to the higher temporal resolution, with good results (median Pearson's Correlation, R, of 0.61 and 0.64 for the parameterized and calibrated SM2RAIN). When 10 days accumulated rainfall is considered, S1-RT1 derived rainfall from the parameterized (calibrated)
SM2RAIN performs quite well, with a median R of 0.61 (0.65), but ASCAT performances are higher, being the median R equal to 0.71 (0.75). At higher temporal steps the performance differences reduce, until ASCAT and S1-RT1 derived rainfall reach almost equal R for the 30 days accumulated rainfall (around 0.75). Similar conclusion can be deduced analysing RMSE index, while for BIAS index the differences between the calibrated and the parameterized SM2RAIN results are larger, probably due to the low spatial resolution of the product used to obtain the Po River Basin climatology, ERA5-Land.
Even if averagely the rainfall from ASCAT seems to be slight better performing than the one from S1, the analysis of the spatial distribution of R shows instead the true benefits of the high resolution SM. In the complex mountain regions, S1 obtains extreme good performance over the valleys and bad performance over the ridges, unsuited for SM remote sensing, whereas ASCAT R is averaged due to the overlaps of the two signals forced by ASCAT lower spatial resolution. S1 derived rainfall is generally better performing than the one from ASCAT also in the northern section of the Po Valley plain, while the latter is
better in the southern section, where irrigation is widely practiced. The fragmentary nature of the irrigation in this area could be the cause of this phenomena: S1-RT1 should be more sensitive than ASCAT to the signal generated by various small fields,





where irrigation in not concurrent, thanks to its higher spatial resolution, but since irrigation is not considered in the benchmark product, the resulting R is reduced.

Summing up, high resolution rainfall from satellite remote sensing is possible and is able to observe features that are averaged
in products with lower spatial resolution, like the precipitation within mountain valleys and potentially the fields' irrigation. Notwithstanding this, the low temporal resolution is currently a limitation for its application in many fields, even if high spatial resolution rainfall at monthly temporal resolution is still useful for fields such as agriculture, water resource management and index-based insurance. Future research steps should try to address this issue, e.g., by exploiting the integration of high spatial resolution products (characterized by low frequency) with high temporal resolution products (characterized by low spatial
resolution).

**Appendix – A**

In some areas, the accuracy of the rainfall obtained from the application of both the calibrated and parameterized SM2RAIN to ASCAT or S1-RT1 SM products is stably low, as discussed in section 4.2. This issue can depend by multiple factors, as SM signal quality, failure of the model or accuracy of the benchmark rainfall product. In this appendix, an attempt to identify those
area is made, by highlighting the pixels in which the Pearson's correlation between the 30 days accumulated rainfall from MCM and the four SM2RAIN derived products is always less than a threshold, fixed at 0.65, as shown in Fig. A-1. Multiple areas of stable low performances can be distinguished in the figure, highlighted in blue. Two main reasons of this behavior can be identified: issues with the SM sensing and issues with the benchmark product.

In particular, the blue areas located in mountainous region in Fig. A-1, in the North and the West of the map, should be affected
by both the source of error, since on orographically complex areas SM retrieval is difficult and weather radar accuracy drops. The areas within the light blue rectangles, are instead characterized by the presence of paddies and water bodies: here the low performance should be caused by low SM quality, due to the impossibility of retrieve SM information over flooded areas. Finally, the remnant blue regions should be affected by low quality of the benchmark product. This can be related either to "bad" performing gauge stations, recognizable through the central position of a gauge with respect to the low performing area
(e.g. the two regions in the Center-East black rectangles), or to issues with weather radar measurements, where the blue patterns are concentrated between two or more raingauges (e.g. the region within the black rectangles on the South-West).

In order to better analyze this aspect, three stations located in within the three black rectangles in Fig. A-1 were selected, together with the nearest neighbor stations. The MCM timeseries of the pixels that includes the stations were extracted, in order to compare them and assess the quality of the considered raingauges. The qualitative comparison of the stations is shown
in Fig. A-2, where the scatter plots for each pair of raingauges is shown together with their position in the map (Fig. A-2a). In particular, a clear issue with the raingauge named A1 can be appreciated in Fig. A-2b, with this sensor measuring rainfall peaks up to 300 mm/day, absent from the nearest gauges. The issue can be confirmed by the low Pearson's correlation between its timeseries and the one of the nearest raingauge, equal to 0.53, that is significantly lower than the mean Pearson's correlation

calculated between each couple of nearest stations within the study area, equal to 0.87 (standard deviation equal to 0.1). Also
Figure A-2c shows strange patterns of rainfall: even if there are no large peaks, several rainfall events are sensed with different
magnitude by the two stations named B1 and B2, as can be noticed by looking at the number of points that tends to the x and
y axis which indicate severe over- or underestimation. Also in this case, the measured Pearson's correlation is lower than the
average, equal to 0.71. Finally, the station C1 measures several peaks of rainfall that are higher than those sensed by the nearest
raingauge, C2. Notwithstanding this, in this case the variation between the two sensors seems to be caused by the natural
spatial variability of the rainfall, as demonstrated by the high Pearson's correlation between the two timeseries, equal to 0.88.
This was expected since the low performing region is not located around one of the stations, but in between them, over a hilly
area that could affect the weather radar measurements

## Appendix – B

In this paper, the performance indexes were calculated at three different temporal steps: 1 day, 10 days and 30 days. In order
to obtain them, the timeseries of each estimated product and of the observed one were accumulated according to the selected
period by considering only the intervals in which the data was available in both the datasets. This choice was made to obtain
the best accurate assessment of each product, by calculating its potential in estimating rainfall against a concurrent dataset.
Notwithstanding this, the comparison of ASCAT and S1-RT1 based on such performances could be biased, because in this
way the analyzed indexes are calculated against two different benchmark datasets, each resulting by the selected product valid
data periods. In this section, we decided therefore to calculate again the performance indexes by accumulating the rainfall of
the observed and estimated datasets only over the periods in which the three datasets (i.e., MCM, ASCAT and S1-RT1) are
available together, and to insert in this appendix the corresponding of Fig. 4 (Fig. B-1) and 5 (Fig. B-2) with the newly
calculated indexes. To further increase the comparison quality and to avoid the period in which just one Sentinel-1 sensor was
in orbit and thus the associated drop in performance, only the data subsequent to 01/10/2016 were considered for the new
indexes calculation.

In comparison with the paper's results, here ASCAT performances increase, due evidently to the removal of some low
performing data, as confirmed by the appearance of some patterns within the ASCAT correlation maps in Fig. B-2a that
resemble the invalid pixel percentage distributions map of Fig. B-2d. Notwithstanding this, the areas in which S1-RT1
outperforms ASCAT are almost confirmed, although shrinked, confirming the paper's results.

## Author contribution

Paolo Filippucci: Conceptualization, Data curation, Formal analysis, Investigation, Methodology, Project administration,
Software, Validation, Visualization, Writing – original draft preparation;
Luca Brocca: Conceptualization, Funding acquisition, Supervision, Writing – review & editing;



Raphael Quast: Data curation, Resources, Writing – review & editing;

Luca Ciabatta: Data curation, Resources, Writing – review & editing;

Carla Saltalippi: Supervision, Writing – review & editing;

Wolfgang Wagner: Supervision, Writing – review & editing;

Angelica Tarpanelli: Supervision, Writing – review & editing.

**Competing interests**

The authors declare that they have no conflict of interest.

**Acknowledgements**

We would like to acknowledge the support of Forschungsförderungsgesellschaft (FFG) Cooperative R&D project through the project DWC Radar (Contract no. CNR-IRPI 24840/2019), the support of European Organisation for the Exploitation of Meteorological Satellites (EUMETSAT) through the project H SAF (Contract no. EUM/C/85/16/LOD, EUM/C/85/16/MIN)

and finally the support of European Space Agency (ESA) through the project DTE Hydrology (Contract no. ESA 4000129870/20/I-NB (CCN N. 1))

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



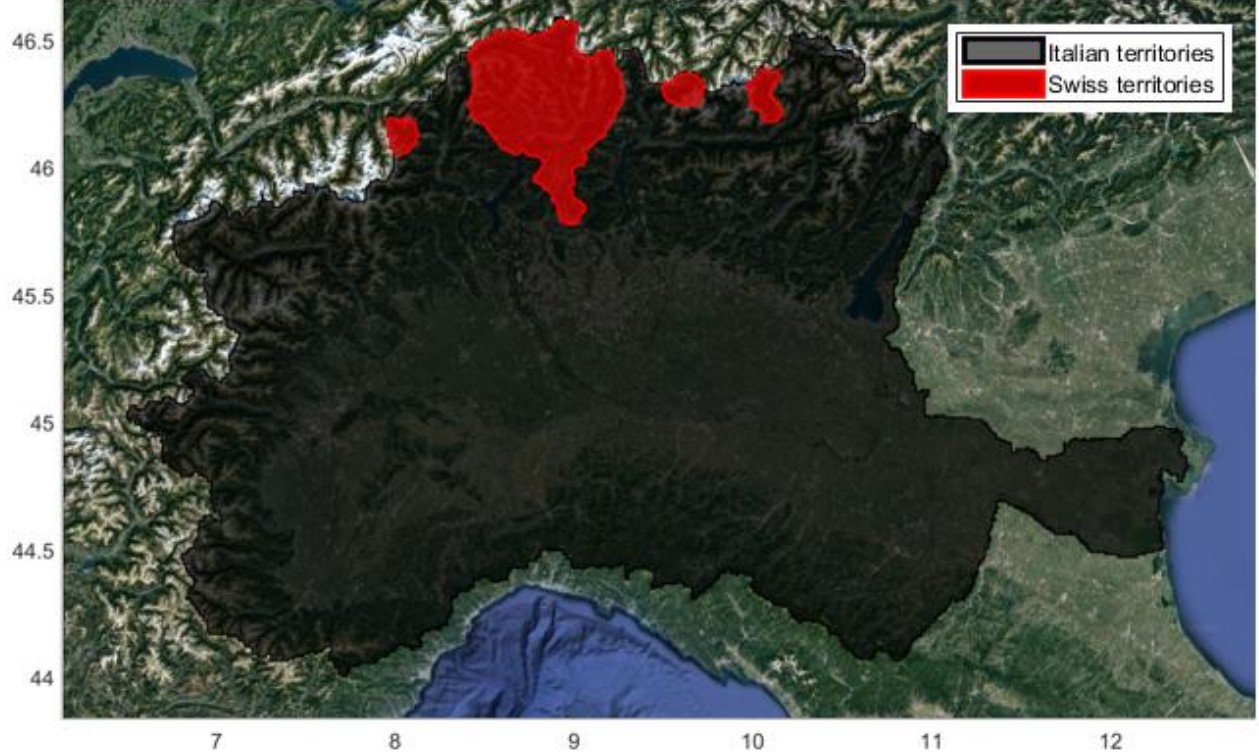

**Figure 1: Po River Basin area. The black shade indicates the study area, inside Italian territory. The Swiss fraction of the basin, excluded from this study, is highlighted in red. Map copyright ©2021 GeoBasis-De/BKG (©2009), Google, Inst. Geogr. Nacional Immagini ©2021 TerraMetrics.**





**Figure 2: Estimated average 30 days rainfall from the parameterized SM2RAIN applied to ASCAT (Panel a) and S1-RT1 (Panel b) SM product for the period 2016-2019. Map copyright ©2021 GeoBasis-De/BKG (©2009), Google, Inst. Geogr. Nacional Immagini ©2021 TerraMetrics.**



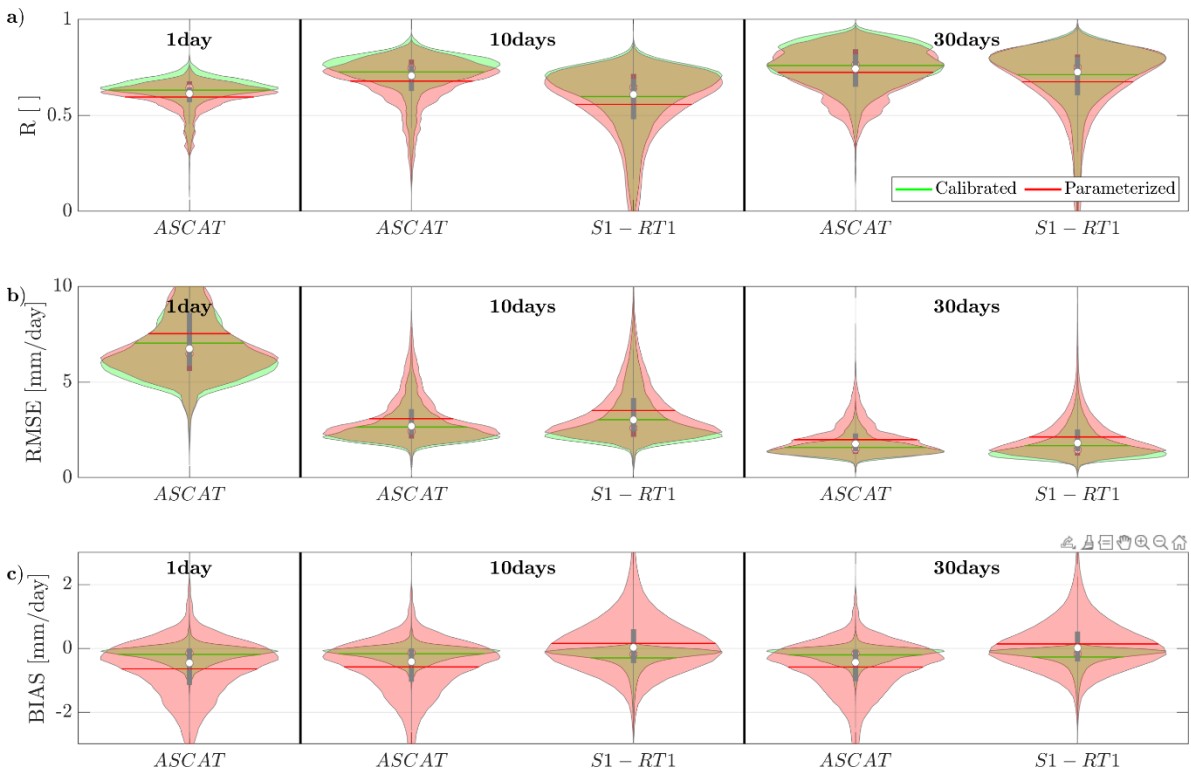

**Figure 3: Violin plots of Pearson Correlation (R, panel a), Root Mean Square Error (RMSE, panel b) and BIAS (panel c) between the rainfall from MCM and from SM2RAIN applied to ASCAT and S1-RT1. ASCAT-derived rainfall was accumulated at 1, 10 and 30 days, while the rainfall from S1-RT1 was accumulated at 10 and 30 days. Each violin shape is obtained by rotating a smoothed kernel density estimator. The green violins are obtained by calibrating SM2RAIN against MCM, while the red violins derived from the parameterized SM2RAIN procedure.**



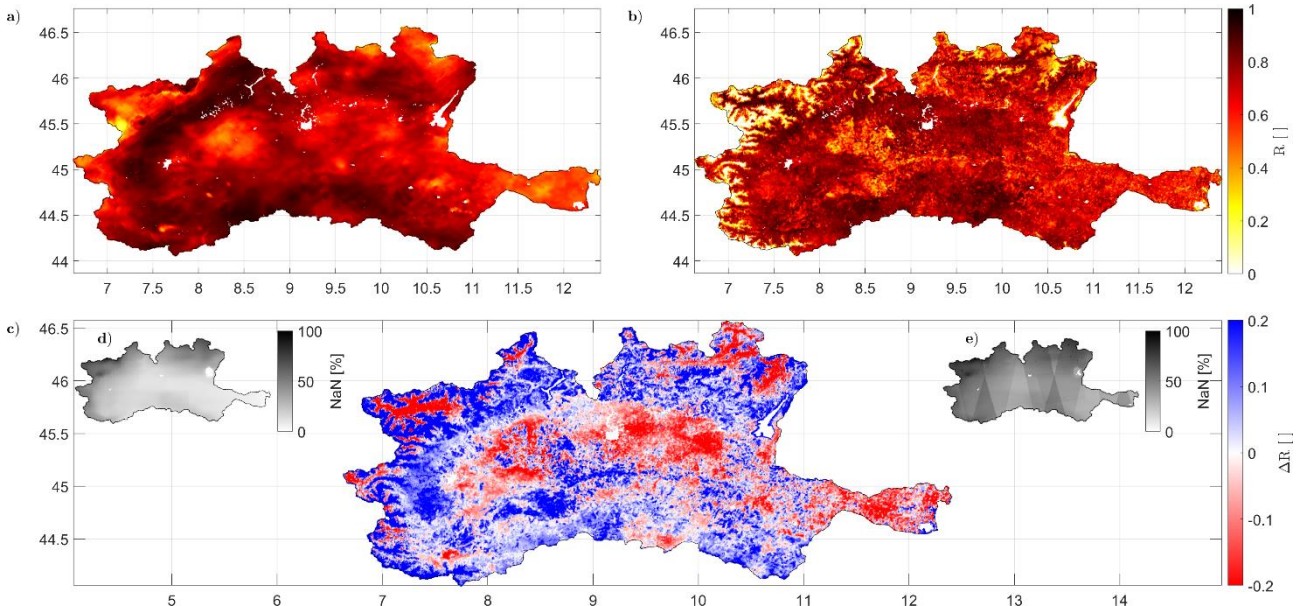

**Figure 4: Spatial Pearson's correlation (R) between the 30 days accumulated rainfall derived from MCM and the application of the parameterized SM2RAIN to ASCAT (panel a) and to S1-RT1 (panel b) SM products. Panel c shows the difference between ASCAT and S1-RT1 correlation maps, while panel d) and e) show the percentage of not valid images per pixel respectively for ASCAT and S1-RT1.**





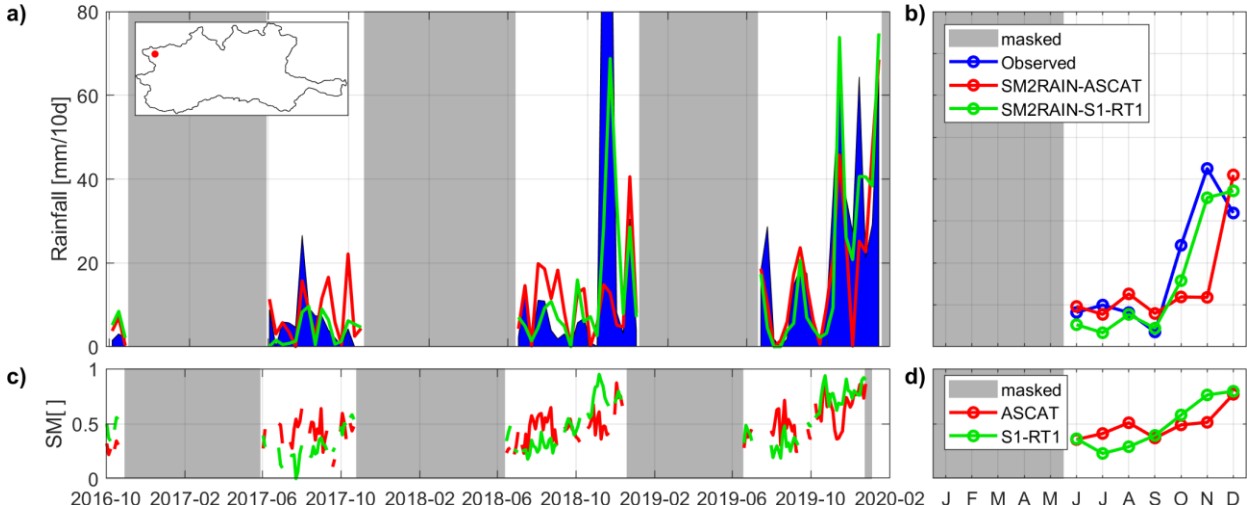

**Figure 5: Example of SM and rainfall timeseries over a pixel (7.152°E, 45.710°N) where the parameterized SM2RAIN applied to S1-RT1 outperforms SM2RAIN-ASCAT. In panel a, the timeseries of the observed (blue) and estimated (red SM2RAIN-ASCAT, green SM2RAIN-S1-RT1) 10-days accumulated rainfall products are shown, while panel c displays SM timeseries averaged with a 3 days window. Finally, panel b and d contain the standard month average of the rainfall and SM products, respectively. The periods masked for frozen soil condition or snow cover are highlighted in grey.**





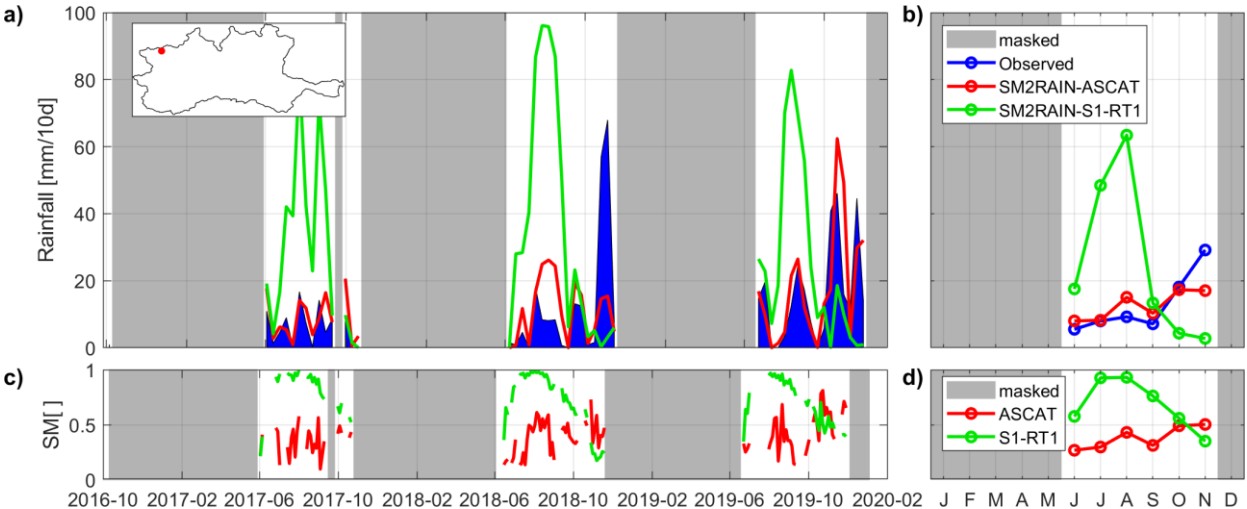

**Figure 6: Example of SM and rainfall timeseries over a pixel (7.410°E, 45.824°N) where the parameterized SM2RAIN-ASCAT outperforms SM2RAIN applied to S1-RT1. In panel a, the timeseries of the observed (blue) and estimated (red SM2RAIN-ASCAT, green SM2RAIN-S1-RT1) 10-days accumulated rainfall products are shown, while panel c displays SM timeseries averaged with a 3 days window. Finally, panel b and d contain the standard month average of the rainfall and SM products, respectively. The periods masked for frozen soil condition or snow cover are highlighted in grey.**



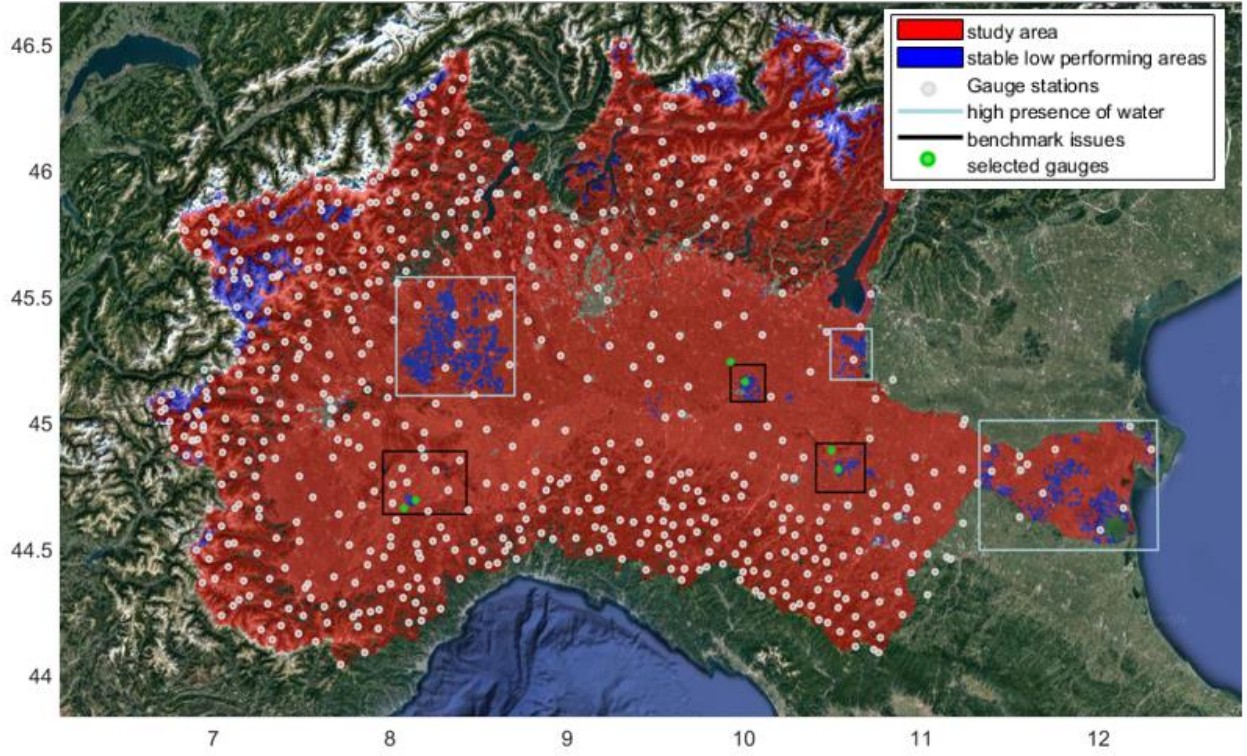

**Figure A-1: Map of Po River basin. The blue pixels indicate the areas where Pearson's correlation between the 30 days accumulated rainfall from MCM and the calibrated and parameterized SM2RAIN applied to ASCAT or S1-RT1 is stably less than a threshold of 0.65. The light blue rectangles surround the areas with paddy areas or abundant water bodies, while black rectangles outline areas with alleged "bad" performing gauge station. Finally, the white dots show the gauge stations location and the green dots the raingauge selected to be further analyzed. Map copyright ©2021 GeoBasis-De/BKG (©2009), Google, Inst. Geogr. Nacional Immagini ©2021 TerraMetrics.**




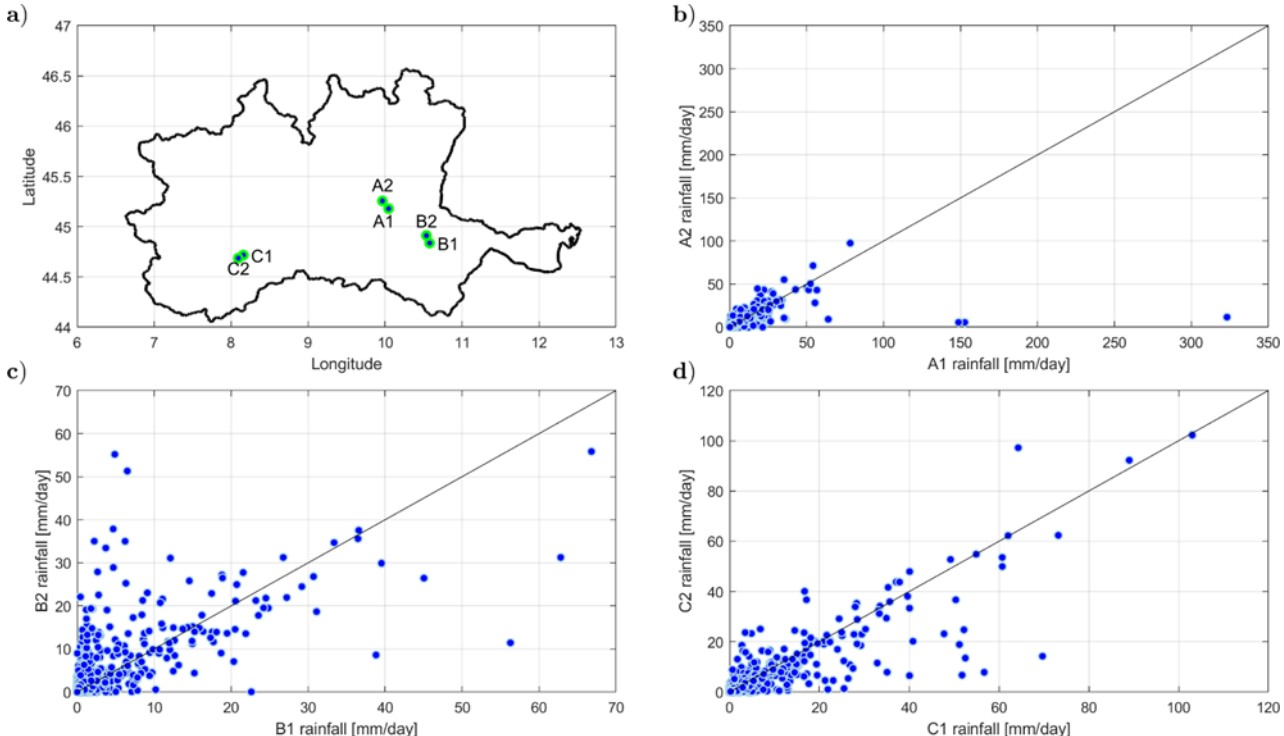

**Figure A-2: Panel a shows the boundary of the Po River basin, together with the position of three couple of stations (A1-A2, B1-B2 and C1-C2) with alleged "bad" MCM performance. The scatter plots of the daily rainfall measured by each couple of stations is then shown in Panel b, c and d.**





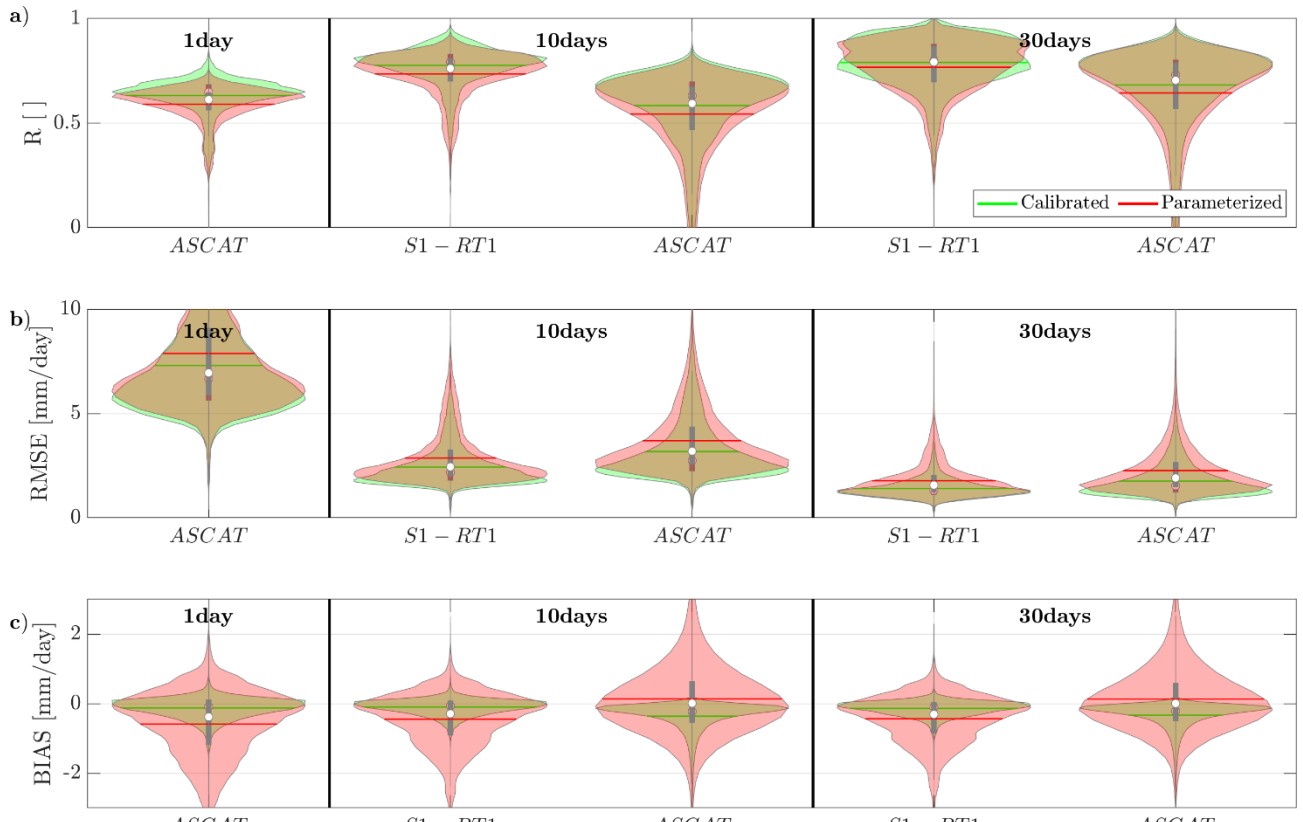

**Figure B-1: Violin plots of Pearson Correlation (R, panel a), Root Mean Square Error (RMSE, panel b) and BIAS (panel c) between the rainfall from MCM and from SM2RAIN applied to ASCAT and S1-RT1. ASCAT-derived rainfall was accumulated at 1, 10 and 30 days, while the rainfall from S1-RT1 was accumulated at 10 and 30 days. Only the periods in which all three products are available**
**are considered in the accumulation. Each violin shape is obtained by rotating a smoothed kernel density estimator. The green violins are obtained by calibrating SM2RAIN against MCM, while the red violins derived from the parameterized SM2RAIN procedure.**



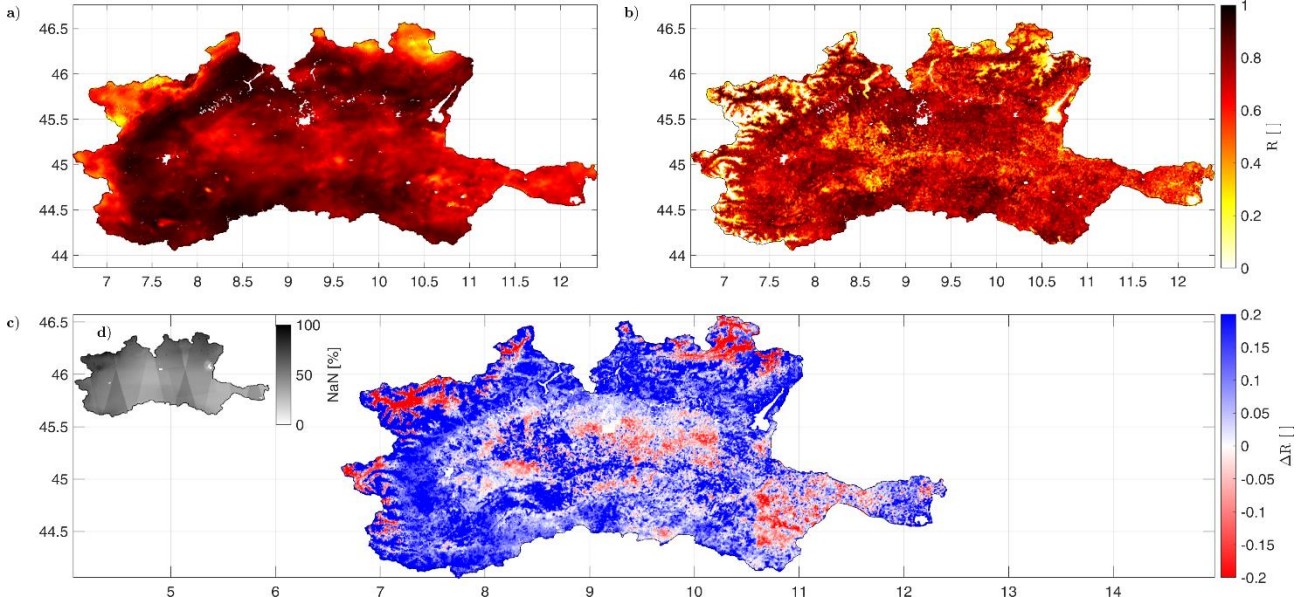

**Figure B-2: Spatial Pearson correlation (R) between the 30 days accumulated rainfall derived from MCM and the application of the parameterized SM2RAIN to ASCAT (panel a) and to S1-RT1 (panel b) SM products, considering only for the periods in which all three products are available. Panel c shows the difference between ASCAT and S1-RT1 correlation maps, while panel d shows the percentage of not valid images per pixel.**