# Peer review of "High resolution (1 km) satellite rainfall estimation from SM2RAIN applied to Sentinel-1: Po River Basin as case study"

_Hydrology and Earth System Sciences, 2021_

## Author Response (AR1)

**Reviewer #1**

This is a very interesting study to explore the Sentinel-1 soil moisture estimates and its potential usefulness in deriving the rainfall estimates at 1km resolution in the Po river basin areas in Italy, using the SM2RAIN algorithm. The study has compared the S1 estimates with ASCAT derived rainfall estimates, and also a gauge+radar derived estimates, with a calibrated version of algorithm and a parameterized version. While this study still presents many challenges in terms of accurately inverse rainfall from SM, especially for S1 with low temporal resolution, and remaining issues with certain geographic region where this algorithm will not apply by nature and parameters are difficult to get, I think the study itself is self-contained and interesting. So I'd suggest moderate to major revisions, before it can move forward.

We thank the reviewer for the valuable suggestions that helped us to clarify and improve the manuscript. A detailed answer to each comment is reported in the sequel.

- Major Comment:

By looking at Fig. 5 & Fig. 6, I cannot help asking what is the usefulness of the SM2RAIN products, because they are highly intermittent, and when they have estimates, the estimates are noisy too (zeros and high values are common). I understand the metrics are adequate (e.g., R>0.6 for subdaily scales), but since it is one of the goals of this study to provide inputs for the hydrologic modeling community for better inputs, I think it would be very critical for the authors to discuss the true usefulness of their estimates to hydrologic modeling, and if still ways to go, to step back on their motivation or concluding remarks.

Fig. 5 and 6 are related to pixels located in a valley inside the mountainous area and on a ridge of the mountain, respectively. The intermittence of the data is due to the frequent frozen soil/snow cover that limits and impact the SM retrieval in the area. Mountainous regions are known to be a challenging environment to obtain reliable satellite SM observations, and hence the estimated rainfall present high levels of noise. Notwithstanding this, the two figures are not intended to make an example of the overall performance of the dataset, rather to add details on the area where SM2RAIN applied to S1-RT1 outperform the one applied to ASCAT. We stressed out this fact in the revised version of the manuscript on Lines 359-361, by stating:

**Since these pixels are selected in a topographic complex area, they should not be considered representatives of the overall performance and availability of the satellite rainfall products, rather an example of the improved performance derived from the use of S1-RT1 high resolution SM with respect to ASCAT.**

We have also added the description of Figure 6 (not included before due to an error in the formatting of the manuscript) and a new figure, in order to show the behaviour of an exemplary pixel selected over the plain (10.684 E° 44.805 N°, with better performance and smaller number of invalid data) and underlined the usefulness of the proposed dataset for the hydrologic modelling community, as suggested (Lines 371-388):

**Figure 6 show instead the timeseries of a pixel selected over the mountain slopes, in the vicinity of the previous one (7.410°E, 45.824°N). While ASCAT SM estimates (Fig. 6c and 6d) show patterns that are similar to those in Fig. 5, S1-RT1 signal is completely different. The SM saturates in the summer period and get dry in autumn, with a strong seasonality that is poorly affected by the rainfall events. This is most probably an issue related to the vegetation-correction, since it adds a strong seasonality to pixels that realistically exhibit little vegetation coverage, also due to the low spatial resolution (with respect to S1-RT1) of the LAI product used for correcting the plants signal. This erroneous seasonality results in a seasonal**

overestimation of SM. As expected, the lower quality of SM observations, greatly affects SM2RAIN capabilities in estimating rainfall in these areas, resulting in very high rainfall rate perceived during summer and very low one during winter, in contrast with the observed data.

Finally, Fig. 7 shows the timeseries of a pixel selected over the plain (10.684°E, 44.805°N). As it can be noted, the period of unavailability of the rainfall data is greatly reduced in comparison with Fig.5 and Fig.6, since this area is characterized by higher temperature during the winter and by lower snow cover probability. Overall, S1-RT1 SM shows a greater variability during the summer season with respect to ASCAT (Fig. 7c-7d), thanks to both the vegetation correction and the higher spatial resolution. This leads to a greater accuracy in the peak rainfall detection of summer 2018 and 2019 (Fig. 7a). In the same panel, it can also be noted an overestimation of 2017 summer rainfall (potentially due to an error in SM estimation or to an irrigation event) and an underestimation of winter 2019 (probably due to SM saturation). Overall, the rainfall estimate from S1-RT1 is in good accordance with the observed one (Fig. 7b), proving both the validity of the derived rainfall product and its usefulness for hydrologic modelling.

[Figure]

Figure 7: Example of SM and rainfall timeseries over a pixel (10.684 E° 44.805 N°) selected in the plain. In panel a, the timeseries of the observed (blue) and estimated (red SM2RAIN-ASCAT, green SM2RAIN-S1-RT1) 10-days accumulated rainfall products are shown, while panel c displays SM timeseries averaged with a 3 days window. Finally, panel b and d contain the standard month average of the rainfall and SM products, respectively. The periods masked for frozen soil condition or snow cover are highlighted in grey.

*Places needing more clarifications:*

L14: it is confusing with "1km resolution" and "500 m spacing" and "25 km spatial resolution (12.5 km spacing)" throughout the paper. Could the authors provide explanations on what the bracket means? Or if it produces similar confusion among other readers, I suggest to remove what's inside the bracket.

These two definitions refer to spatial resolution and spatial sampling: according to Shannon's sampling theorem, in order to preserve the spatial resolution of the original image, the digitizing device must utilize a sampling interval that is no greater than one-half the size of the smallest resolvable feature of the optical image. This is equivalent to acquiring samples at twice the highest spatial frequency contained in the image, a reference point commonly referred to as the Nyquist criterion (https://www.olympus-lifescience.com/en/microscope-resource/primer/java/digitalimaging/processing/samplefrequency/). This approach was used by Wagner et al. (2013), to improve the sampling of the ASCAT product, and repeated

also for the SM product derived from S1. We introduced a brief explanation of the concept in the revised manuscript in Lines 124-126:

**The spatial sampling was fixed at one-half of the spatial resolution, according to the Nyquist-Shannon sampling theorem, to maximize the details of each SM datum (Wagner et al., 2013).**

L55 would require citations because readers would like to know which algorithms the authors are talking about and why SM2RAIN stands out.

We thank the reviewer for its suggestion. The following reference has been added: **Crow et al., 2009; 2011; Pellarin et al., 2013; Wanders et al., 2015.**

**Crow, W. T., Huffman, G. F., Bindlish, R., and Jackson, T. J.: Improving satellite rainfall accumulation estimates using spaceborne soil moisture retrievals, J. Hydrometeorol., 10, 199–212, 2009.**

**Crow, W. T., van den Berg, M. J., Huffman, G. J., and Pellarin, T.: Correcting rainfall using satellite-based surface soil moisture retrievals: The Soil Moisture Analysis Rainfall Tool (SMART), Water Resour. Res., 47, W08521, https://doi.org/10.1029/2011WR010576, 2011.**

**Pellarin, T., Louvet, S., Gruhier, C., Quantin, G., and Legout, C.: A simple and effective method for correcting soil moisture and precipitation estimates using AMSR-E measurements, Remote Sens. Environ., 136, 28–36, 2013.**

**Wanders, N., Pan, M., and Wood, E. F.: Correction of real-time satellite precipitation with multi-sensor satellite observations of land surface variables, Remote Sens. Environ., 160, 206–221, 2015.**

SM2RAIN stands out between them as it is the most used among them, as demonstrated by the more than double number of citations.

L148: I am not sure how HESS handles citing unpublished articles. It seems the details on how S1 data were used to derive the 1km SM product (L134-147) is key to the overall conclusions. I believe in Quast et al., in preparation., authors should have concluded on the SM performances based on their RS processing procedures. I think it would be very helpful if the authors mention some of the major conclusions from Quast et al., if this unpublished paper needs to be here.

In Quast et.al., a description of the retrieval algorithm alongside an extensive analysis of the resulting soil-moisture retrieval performance with respect to ERA5 top-layer soil-moisture (swvl1) will be given. Since a comprehensive analysis of the soil-moisture retrieval performance requires the consideration of numerous influencing factors (model parametrization, topography, landcover, spatio-temporal resolution etc.) including details on the performance-analysis is out of scope for this publication. However, a sentence summarizing the main findings has been added to the paper in lines 156-158:

…Within the retrieval-procedure, a unique value for N is obtained for each timestamp, alongside a temporally constant estimate for ts and an orbit-specific estimate for ω for each pixel individually.  **"A comparison of the obtained RT1 soil-moisture retrievals to ERA5-Land top-layer volumetric water content (swvl1) for a set of ~138 000 pixels over a 4 year time-period from 2016 to 2019 achieves an overall (median) Pearson correlation of 0.55 for areas classified as croplands and 0.65 for areas primarily covered by natural vegetation."** A detailed description and performance-analysis of the used soil-moisture dataset will be given in Quast et al., in preparation. ….

L173-181: it is not clear why ERA-5's precipitation data (and then derived rainfall data) are used to calculate the daily climatology, for the parameterized version of the SM2RAIN rainfall estimates. In my understanding, ERA-5's rainfall data were coarse and also not performing the best for Italy. Later, the authors also discussed the high bias may be also associated with the ERA-5 (Lin 278). Have the authors tried to use other rainfall estimates (with high spatial resolution and better fidelity in your study region) for this?

We have selected ERA5 due to its global availability and long-term coverage. We compared the downscaled product against the benchmark rainfall, obtaining average good correlation (0.701). We have also tried different products. The most promising one was CHELSAv1, a rainfall climatology product characterized nominally by around 1 km spatial resolution. Notwithstanding this, we found that the product was heavily driven by topography in the selected area, and that better performances could be obtained by using ERA5 rainfall.

L194: how was the soil porosity derived? Which soil texture data was used?

The soil porosity was not derived. Both the parameterized and calibrated product estimate directly the parameter Z* that already comprehend the soil porosity, as specified by Lines 215 and 224.

Fig. 1: this is not a very typical figure commonly seen in an academic journal. I suggest the author to draw the Po river basin boundary, and overlay it with the country boundary maps, on top of topographic maps. This way, the information should be more clearly conveyed. The later analyses part can also have better reference information to the topography here (for example, I suggest when discussing about results in Fig. 4, topo in Fig. 1 can be used as a reference).

We thank the reviewer for its suggestion. Figure 1 has been changed with the following one:

[Figure]

Also, the reference to the figure has been adjusted as following (Lines 114-115):

…In this study, the  fraction of the Po River basin **external from the Italian boundaries (** **black line** in Fig. 1) was excluded from the analysis due to the unavailability of raingauge data.

Finally, the following sentence has been added to the manuscript in Line 326, as suggested:

First of all, it should be noted that while ASCAT derived rainfall product shows average correlation values over the mountainous region in the North and West of the map **(see Fig.1 for comparison with the DEM map)**

Fig. 2: figure caption: "30 days rainfall" should be which 30 days or which month?

The figure caption has been corrected. Now it states:

Figure 2: Estimated average 30 days **accumulated** rainfall from the parameterized SM2RAIN applied to ASCAT (Panel a) and S1-RT1 (Panel b) SM product for the period 2016-2019.

- Minor style issues:

many acronyms have been defined multiple times such as DEM, MCM, etc. I suggest the authors to check them throughout the paper and remove duplicate acronyms (only define them at the first appearance).

The double definition of DEM in line 249 has been removed. Each other acronym was defined twice, once in the text and once in the Conclusion, to facilitate the reading. In order to avoid misunderstanding, we removed the definitions in the Conclusion.

L291: note inconsistent citation style

We revised the text as suggested.

L365: "slight" to "slightly"

We revised the text as suggested.

L367: "extreme" to "extremely"

We revised the text as suggested.

As also a non-native speaker, I found this paper difficult to follow at many places because of the use of non-native English. I suggest this paper to go through review or editorial processes with native speakers, before its final publication.

We thank the reviewer for its suggestion. We have reviewed the text again and we decreased the non-standard usages, as suggested.

**Reviewer #2**

The paper titled "High resolution (1 km) satellite rainfall estimation from SM2RAIN applied to Sentinel-1: Po River Basin as case study" explores how precipitation estimates can be retrieved by inverting the precipitation necessary to produce the soil moisture signal measured in both ASCAT and S1. They compare their results between the 12 km ASCAT product and the target 1 km product. The end result shows that the overall performance of the higher resolution soil moisture product is mostly similar to the coarser resolution ASCAT one. However, there are large difference in performance depending on the topographic/land cover domain. Overall, the study is well presented and relevant to the community. I recommend the paper be accepted after moderate revisions.

We thank the reviewer for the valuable suggestions that helped us to clarify and improve the manuscript. A detailed answer to each comment is reported in the sequel.

My main comment is that the paper would be well served by a large enhancement of the discussion. The paper shows how the finer detailed SM1-derived precipitation estimates are at best equal to the ASCAT product which at first glance is a discouraging result. However, the correlation maps tell us a much more interesting story. As is noted in the paper, valley vs peaks seems to be playing an important role in the retrieval accuracy. The authors then present arguments of why this might be the case but I believe these arguments could be further developed. Here are some relevant questions that could be discussed accordingly.

1. What could be done moving forward to address the weaknesses observed over these regions? What are the reasons for these weaknesses?

2. What other precipitation datasets could be used to evaluate the product? Since there are large errors over the high mountainous regions, this is most likely strongly driven by errors in the precipitation product. I believe that the use of an algorithm such as that used in PRISM over the US would be interesting to improve these estimates.

3. How much of the error is attributed to the SM2RAIN model parameters and how much to the noise of the retrieval?

I am not suggesting that the authors solve these issues in this paper, but I do think a comprehensive discussion around these ideas would be very useful. In many ways, the appendices already include many of these ideas; when creating a discussion section, I would suggest scavenging from these sections and then eliminating the appendices. The paper could handle more figures, so some in the appendices could be added in here.

We thank the reviewer for their suggestion, we agree with it. We added a "Discussion" Section (Lines 389—437) using as the basis the appendix A, appropriately modified to address the highlighted issues. The main modifications to the text are reported in the follow:

[revised manuscript text omitted]

The following sentence has been also added to conclusion (Lines 471-474), to underline the results obtained in this section:

**Some areas with stable low performance of rainfall estimation were also identified (Fig. 8), caused by the limitations of SM2RAIN algorithm (e.g., areas in which runoff rate is not negligible), of the SM remote sensing (areas in which SM estimation is impossible, e.g., flooded or snow covered areas) and of the benchmark product (e.g., topographically complex areas).**

---

## Author Response (AR2)

**Editor**

Dear authors,

Thank you for taking your time to revise the work. Your revisions were seen by the two referees - the first one appears to be satisfied with the revision, while the second one still have some concerns regarding the clarity of the contents. The referee has provided suggestions to further improve the work. Therefore I would like to ask you to revise the manuscript one more time, taking care of remaining referee's concerns.

I look forward to your revisions.

Best regards,
Rohini Kumar

We thank the editor for the comments. Below a specific reply to each comment of the reviewer is inserted.

**Reviewer #1**

I think the authors have mostly addressed my earlier comments, although I still have some concerns on the clarity of this MS. Below is some suggestions to improve the delivery of this paper.

We thank the reviewer for the valuable suggestions that helped us to clarify and improve the manuscript. A detailed answer to each comment is reported in the sequel.

Style issues:
Line 21: Incorrect use of "to" after "since".

The sentence was corrected.

Line 60: "of" is supposed to come after "regardless"

The sentence was corrected.

Line 70: "new" to "newly"

The sentence was corrected.

Line138: "-" after soil not needed.

The sentence was corrected.

Line 141: "for" not needed.

The sentence was corrected.

Line 261-262: Fig. 4e comes before Fig. 3 in the article.

The reference to Figure 4e was removed and placed after Fig.3 is introduced, in Line 335-336.

"Notwithstanding this, the absence of patterns in the maps that resemble the NaN distribution percentage shown in Fig. 4d and 4e, fosters the validity of the analysis, **even if S1 temporal resolution still affects the average rainfall pattern (compare Fig. 4e with Fig. 2b)**."

Line 348: "equal" to "equally".

This error is referred to the first version of the manuscript. It was already corrected in the last revision.

Line 352: Preposition word missing after "related".

This error is referred to the first version of the manuscript. It was already corrected in the last revision.

Line 356: "it" missing before "is needed".

This error is referred to the first version of the manuscript. It was already corrected in the last revision.

Other suggestions:
Line 230: The descriptions of the metrics (R,BIAS,RMSE) can be much more comprehensible if the authors could provide the corresponding equations.

The equations have been added to the description.

In the Conclusion section, it is recommended that the authors provide more general and clearer descriptions of their results, while detailed explanations can be addressed in the Results section (In short, conclusions are not concise enough). Moreover, grammatic issues in some sentences undermine the conveyance of the analyses (especially the first two paragraphs of the section), which needs revision.

According to the line numbers provided by the reviewer and to their request of correct style issues that were already been corrected in the last revision (specifically the ones in line 348,352 and 356 of the first version of the manuscript), we believe that the reviewer erroneously analyzed the first version of the manuscript and not the revised version. The grammatic issues and the descriptions of the results they are referring to, have already been corrected in the last review.

Line 275: It seems the spatial resolution and average rainfall pattern of the ERA5-Land data significantly influence these discussed metrics. Whether there exist other choices in all datasets is unclear. It is recommended that the authors provide a brief discussion here.

We have selected ERA5 due to its global availability and long-term coverage. We compared the downscaled product against the benchmark rainfall, obtaining average good correlation (0.701). We have also tried different products. The most promising one was CHELSAv1, a rainfall climatology product characterized nominally by around 1 km spatial resolution. Notwithstanding this, we found that the product was heavily driven by topography in the selected area, and that better performances could be obtained by using ERA5 rainfall. A sentence was added to underline the reasons of ERA5 selection at line 187:

"**This product was selected due to the high temporal coverage, its worldwide availability and its accuracy.**"